# Gilteritinib: The Story of a Proceeding Success into Hard-to-Treat *FLT3*-Mutated AML Patients

**DOI:** 10.3390/jcm12113647

**Published:** 2023-05-24

**Authors:** Matteo Molica, Salvatore Perrone, Marco Rossi

**Affiliations:** 1Department of Hematology-Oncology, Azienda Ospedaliera Pugliese-Ciaccio, 88100 Catanzaro, Italy; rossim@unicz.it; 2Department of Hematology, Polo Universitario Pontino, S.M. Goretti Hospital, 04100 Latina, Italy; sperrone@hotmail.it

**Keywords:** *FLT3* mutations, resistant/relapsed acute myeloid leukemia, tyrosine kinase inhibitors, gilteritinib

## Abstract

The traditionally dismal outcome of acute myeloid leukemia (AML) patients carrying the FMS-related tyrosine kinase 3 (*FLT3*) mutations has been mitigated by the recent introduction of tyrosine kinase inhibitors (TKI) into clinics, such as midostaurin and gilteritinib. The present work summarizes the clinical data that led to the use of gilteritinib in clinical practice. Gilteritinib is a second-generation TKI with deeper single-agent activity than first-generation drugs against both *FLT3–ITD* and TKD mutations in human studies. Moreover, the phase I/II dose-escalation, dose-expansion Chrysalis trial showed an acceptable safety profile of gilteritinib (diarrhea, elevated aspartate aminotransferase, febrile neutropenia, anemia, thrombocytopenia, sepsis, and pneumonia) and a 49% overall response rate (ORR) in 191 *FLT3*-mutated relapsed/refractory (R/R) AML patients. In 2019, the pivotal ADMIRAL trial showed that the median overall survival was significantly longer in patients treated with gilteritinib than among those receiving chemotherapy (9.3 vs. 5.6 months, respectively) and the ORR to gilteritinib was 67.6%, outperforming the 25.8% for chemotherapy arm and leading to the license for its clinical use by the US Food and Drug Administration. Since then, several real-world experiences have confirmed the positive results in the R/R AML setting. Finally, gilteritinib-based combinations currently under investigation, with several compounds (venetoclax, azacitidine, conventional chemotherapy, etc.) and some practical tips (maintenance after allogeneic transplantation, interaction with antifungal drugs, extramedullary disease, and onset of resistance), will be analyzed in detail in this review.

## 1. Introduction

It is now widely accepted that the class III receptor tyrosine kinase *FLT3* mutation status distinguishes a subtype of acute myeloid leukemia (AML) with a poor prognosis. Indeed, *FLT3*-mutated AMLs retain higher relapse rates and shorter remission duration following initial therapy (6 months vs. 11.5 months for those without *FLT3* internal tandem duplication (ITD) mutations), as well as reduced disease-free survival (16% to 27% vs. 41% at 5 years) and overall survival (OS) (15% to 31% vs. 42% at 5 years) [1,2,3]. Relapsed/refractory (R/R) AML has a median OS of 4–7 months with standard chemotherapy approaches [4,5,6,7], emphasizing the importance of newly approved targeted therapies and the need for additional treatment options. *FLT3*/ITD and *FLT3*/TKD mutations are ideal targets for small molecule inhibitors. On 21 September 2018, gilteritinib was approved in Japan for the treatment of R/R *FLT3*-mutated AML; on November 28th of the same year, the US Food and Drug Administration (FDA) also declared marketing approval of gilteritinib for the same indication as in the United States. In the phase III ADMIRAL study, gilteritinib considerably outperformed salvage chemotherapy in terms of OS and the response rate for complete remission with full or partial hematological recovery maintaining a manageable toxicity profile [8]. Furthermore, gilteritinib represented a valid treatment approach as bridge to transplant in this critical subgroup of AML [8]. Overall, these data have led to a therapy shift into the AML treatment scenario, establishing gilteritinib as the new gold standard for R/R *FLT3*-mutated AML, and incorporation of the drug into frontline regimens will likely become the standard therapeutic strategy for de novo *FLT3*-mutated AML. In the future, novel combination approaches promise to further revolutionize the therapeutic landscape of this AML setting. This review will discuss clinical trials and real-life studies’ data of gilteritinib in R/R AML and as maintenance approach after transplant and explore alternative combinations with chemotherapy or other small molecules in de novo and R/R AML.

## 2. Pharmacodynamics and Pharmacokinetics of Gilteritinib

Gilteritinib is a next-generation tyrosine kinase inhibitor (TKI) primarily targeting *FLT3* and *AXL* (an onco-genic tyrosine kinase) receptors [9]. Median maximum concentration is reached after 2–6 h following single and repeat dosing of oral gilteritinib (rapid absorption with or without food); mean elimination half-life was 113 h. Elimination was primarily via feces. Gilteritinib is primarily metabolized via cytochrome CYP3A4; coadministration of gilteritinib with itraconazole (a strong P-glycoprotein inhibitor and CYP3A4 inhibitor) or rifampicin (a strong P-glycoprotein inducer and CYP3A inducer) can significantly interfere with its pharmacokinetic profile [10]. Compared with first-generation multitargeted TKIs, it is more selective to *FLT3* and has greater potency. It blocks *FLT3* receptors’ ATP-binding site competitively, thus inhibiting receptor signaling and halting cell cycle [11]. Cellular experiments have shown powerful inhibitory effects on *FLT3* mutations (*FLT3–ITD* and *FLT3*-D835Y point mutations in particular) [12]. Since both *FLT3–ITD* and *FLT3–TKD* mutations promote constitutive *FLT3* kinase activity, sustaining leukemic cell proliferation and survival, gilteritinib-mediated inhibitory effects have the potential to lessen the leukemia burden of AML patients (Figure 1). It is classified as a type I inhibitor, generally unaffected by mutations in the activation loop (e.g., at D835) [13]. Moreover, gilteritinib promotes apoptosis in *FLT3–ITD* mutations carrying tumor cells in vitro [9]. In xenografted mice models, oral administration of gilteritinib lowered phosphorylated *FLT3* levels by 40% after 1 h [12], while a single dosage was sufficient to reduce the phosphorylation of STAT-5, a known downstream *FLT3* target [12]. Following successive 120 mg doses of gilteritinib in patients with R/R-AML, approximately 90% of *FLT3* phosphorylation was decreased, with inhibition starting to take place 24 h after the first dosage [9]. When oral gilteritinib (1–10 mg/kg) was given to mice once every day for 28 days, tumor development was significantly suppressed by 63–100% (*p* = 0.05) [12]. Although gilteritinib did not influence the in vitro reduction in tumor growth or induction of apoptosis, stimulation of the *FLT3* ligand can raise the chance of resistance to other *FLT3* inhibitors [14]. Given that *AXL* activation is a known resistance mechanism to *FLT3* inhibitors and that *AXL* inhibition can slow the growth of *FLT3–ITD* AML tumors, gilteritinib additional activity against *AXL* may also be advantageous [15]. In comparison with other less specific TKIs, gilteritinib may present a lower clinical risk of side events, such as myelosuppression [12]. Inhibition of *c-KIT* (an oncogene encoding *KIT*, a platelet-derived growth factor receptor essential for hematopoiesis) is expected to provoke severe myelosuppressive effects because *FLT3* and *KIT* structures are remarkably similar [10]. Thus, the risk of myelosuppression with gilteritinib is anticipated to be lower than with other TKIs because it has no impact on *c-KIT* [10]. Based on in vitro findings, CYP3A4 primarily metabolizes gilteritinib [10]. The main metabolites identified in animal investigations are M17, M16, and M10 (all accounting for less than 10% of the parent exposure); it is unknown if these metabolites have any effect on *FLT3* or *AXL* receptors [9]. Since gilteritinib is a P-glycoprotein (P-gp) substrate, a multidrug transporter that actively pumps substances out of the cell and away from their target regions [16], it might exert an inhibitory effect on BCRP, P-gp, and OCT1 in the small intestine as well as the liver [9]. In vivo, gilteritinib neither induces nor inhibits CYP3A4 or MATE1. Since gilteritinib may decrease the effectiveness of 5-HT2B or sigma nonspecific receptor targeting medications in vitro (such as escitalopram), it should only be used in rare conditions together with these medications [9]. Reduced gilteritinib plasma concentrations are caused by coadministration with a P-gp and potent CYP3A inducer, hence this should be avoided [9]. Conversely, gilteritinib exposure is increased when it is administered concurrently with a potent CYP3A and/or P-gp inhibitor [10]. For instance, coadministration of a single 10 mg dose of gilteritinib with 200 mg of itraconazole per day for 28 days raised Cmax and AUC in healthy individuals by 20% and 120%, respectively [9]. A concurrent strong CYP3A and/or P-gp inhibitor increased exposure in individuals with R/R-AML by about 1.5 times [9].

## 3. Clinical Trials Including Gilteritinib as Monotherapy

### 3.1. Chrysalis Trial

Gilteritinib was evaluated for its safety, pharmacokinetics, pharmacodynamics, and antileukemic activities in this first-in-human, open-label phase I/II dose-escalation, dose-expansion Chrysalis trial (NCT02014558) in patients with R/R AML. This study included patients with wild-type (wt) *FLT3* (*n* = 58) and *FLT3* mutation (*n* = 191), totaling 252 R/R AML patients. Participants were assigned to receive a once-daily oral dose of gilteritinib ranging from 20 mg to 450 mg and were enrolled in one of seven dose-escalation (*n* = 23) or dose-expansion (*n* = 229) groups. Overall, gilteritinib was well-tolerated; the maximum tolerated dose (MTD) was established at 300 mg/day, when two out of three patients enrolled in the 450 mg dose-escalation cohort had two dose-limiting toxicities (grade 3 diarrhea and grade 3 elevated aspartate aminotransferase). Most frequent grade 3–4 adverse events (AEs) included febrile neutropenia (39%), anemia (24%), thrombocytopenia (13%), sepsis (11%), and pneumonia (11%); death occurred in ninety-five patients, with seven deaths judged possibly or probably related to treatment. At least 90% of *FLT3* phosphorylation inhibition was observed by day 8 in most patients receiving a daily dose of ≥80 mg. Overall response rate (ORR) in the entire population was 40%; ORR in *FLT3*-mutated (*n* = 191) and *FLT3*wt (*n* = 58) patients was 49% and 12%, respectively. Remarkably, the ORR was enhanced in *FLT3*-mutated patients at doses ≥ 80 mg/day, resulting in 52%. The median OS in the two subgroups was 30 and 17 weeks, respectively [17]. In *FLT3*-mutated patients with R/R AML, gilteritinib monotherapy was well-tolerated and produced frequent and persistent clinical responses. In *FLT3*-mutated patients treated at levels that consistently and potently suppressed *FLT3* phosphorylation, antileukemic responses were enhanced. Gilteritinib related poor efficacy in patients without *FLT3* mutations, suggesting that this approach is very selective by its activity against *FLT3*.

### 3.2. ADMIRAL Trial

The phase III ADMIRAL trial showing improved OS with gilteritinib vs. salvage chemotherapy in patients with R/R FLT3-mutated AML led to the FDA approval of the drug in this setting. A total of 371 patients from 14 different countries were randomly enrolled 2:1 to receive gilteritinib at 120 mg/die (*n* = 124) or investigator’s choice of salvage chemotherapy (MEC, FLAG-IDA, low-dose cytarabine, or azacitidine) (*n* = 124) with cycles of 28 days. ITD-FLT3 mutation was detected in 87% and 91% in the gilteritinib and salvage chemotherapy group, respectively, and the preselected salvage chemotherapy was a high-intensity regimen in 60% of both groups. After a median follow-up of 17.8 months, the median OS was 9.3 vs. 5.6 months (*p* < 0.001) in the two subgroups, respectively, with the benefit of gilteritinib maintained also in analysis-censoring survival data at the time of allogenic hematopoietic stem cell transplantation (HSCT). The rate of complete remission (CR) with full or partial hematologic recovery in the two groups was 34% vs. 15.3% (absolute 18.6% risk difference), with a median duration of response in the gilteritinib group of 11 months. The median event-free survival (EFS) was significantly different between the two subcategories (2.8 vs. 0.7 months; HR 0.79, 95% CI = 0.58–1.09) [8]. Recently, a 2-year follow-up of the ADMIRAL trial after the primary analysis was reported to clarify the long-term treatment effects and safety of gilteritinib in *FLT3*-mutated R/R AML. The 2-year estimated survival rates were 20.6% and 14.2% in the gilteritinib and salvage chemotherapy groups; the survival benefit of gilteritinib was maintained in the *FLT3–ITD* mutation subgroup and in patients with a high *FLT3–ITD* allelic ratio, while it was not observed in the *FLT3–TKD* subgroup or in patients with a low *FLT3–ITD* allelic ratio. The 2-year cumulative relapse rates in gilteritinib-treated patients who achieved a CR or composite CR were 52.6% and 75.7%, respectively. In total, 26 patients treated with gilteritinib were still alive after 2 years of treatment without relapsing; among them, 18 underwent HSCT and 16 received gilteritinib after transplant. In this setting, most patients were aged <65 years (84.6%), treated with high-intensity treatment before randomization (76.9%), and had not received previous *FLT3* inhibitors (96.1%). The most common reported AEs during the first and second year of treatment were the increased levels of transaminases. Compared with the first year of gilteritinib therapy, in the second year, a reduced incidence of these AEs was observed [18]. These data confirmed the long-term benefit of gilteritinib treatment either in patients who did not undergo transplant or in patients who continued gilteritinib in the post-transplant phase. Recently, Smith et al. analyzed the molecular profile of R/R AML patients enrolled in the ADMIRAL trial focusing on the potential relationship between comutations in molecular partners of *FLT3* and response to treatment [19]. At the time of enrollment, patients were classified in the following subgroups: DNA methylation/hydroxymethylation (41.2%), transcription factors/regulators (26.3%), chromatin–spliceosome–other (17.4%), receptor tyrosine kinase (RTK)–Ras signaling (7.8%), *TP53*-aneuploidy (3.6%), *NPM1* (47.9%), *DNMT3A* (31.9%), *DNMT3A*/*NPM1* (23.8%), *WT1* (18.0%), and *IDH1*/*IDH2* (15.5%). Response rates before HSCT appeared higher in the gilteritinib arm vs. the standard chemotherapy arm across all gene categories except *TP53*-aneuploidy, which included a small series of patients (*n* = 13). Longer survival was identified among *NPM1*-mutated, DNA methylation/hydroxymethylation, and transcription factor categories, as well as in comutated *DNMT3A*, *WT1*, and dual-mutated *DNMT3A* and *NPM1* gene categories in the gilteritinib arm as compared with the standard chemotherapy arm. Patients with *DNMT3A*/*NPM1* comutations treated with gilteritinib showed the most favorable outcomes compared with all the others molecular subgroups. Furthermore, OS results observed with gilteritinib were not negatively impacted by *FLT3–ITD* allelic ratio, *FLT3–ITD* length, or multiple *FLT3–ITD* mutations. In the subgroup of patients with *FLT3–ITD* lengths >51 bp, the median OS was 10.4 vs. 6.0 months in the gilteritinib arm and in the standard chemotherapy arm (HR = 0.480; 95% CI, 0.311–0.742), while among patients who presented at baseline with multiple *FLT3–ITD* mutations, median OS was 8.3 months and 3.5 months, respectively (HR = 0.624; 95% CI, 0.331–1.175). In addition, patients with a high *FLT3–ITD* allelic ratio (≥0.77) who received gilteritinib showed a significantly longer OS (7.1 vs. 4.3 months; HR = 0.49; 95% CI, 0.34–0.71). Of the 247 gilteritinib-treated patients, relapse was observed in 75 patients (30%) who had achieved any type of CR; among them, 40 (53.3%) had blood or bone marrow samples available for analysis at baseline and relapse. Overall, 27 out of the 40 relapsed patients (67.5%) had developed new gene mutations during gilteritinib therapy. New mutations in Ras/MAPK pathway genes were detected in 18 patients at the time of relapse, with the most frequently mutated Ras/MAPK pathway genes including *NRAS* (61.1%), *PTPN11* (44.4%), and *KRAS* (38.9%); however, the presence of Ras/MAPK pathway gene mutations at baseline did not affect a potential response to gilteritinib (the rate of composite CR before HSCT in gilteritinib-treated patients with Ras/MAPK pathway gene mutations at baseline was 33.3%) [19]. These results shed light on the molecular profile of *FLT3*-mutated R/R AML, the effect of *FLT3* inhibitors on mutational evolution, linked to treatment resistance, and the efficacy of gilteritinib across a broad range of molecular and genetic subgroups.

Perl et al. [20] retrospectively compared clinical outcomes of patients enrolled in the CHRYSALIS and ADMIRAL trials who had received prior midostaurin or sorafenib against those without prior *FLT3* tyrosine kinase inhibitor (TKI) exposure. Patients who received a *FLT3* TKI prior to gilteritinib (CHRYSALIS, 42%; ADMIRAL, 52%) and those who did not (CHRYSALIS, 43%; ADMIRAL, 55%) both showed high rates of composite complete remission (CRc). In the ADMIRAL trial, among patients who had previously received a *FLT3* TKI, the gilteritinib arm had a higher CRc rate (52%) and a tendency toward a longer median OS than the standard chemotherapy arm (CRc = 20%; overall survival, 5.1 months; HR = 0.602; 95% CI: 0.299, 1.210). With prior *FLT3* TKI exposure, the duration of remission was shorter [20]. These results also establish gilteritinib as a valid treatment option for patients with *FLT3*-mutated R/R AML who had previously received sorafenib or midostaurin. Table 1 summarizes the most significant efficacy and outcomes data of ADMIRAL trial.

## 4. Real-Life Experiences with Gilteritinib in R/R AML

The French AML Intergroup ALFA/FILO retrospectively analyzed a real-world series of R/R *FLT3*-mutated AML patients (*n* = 167) treated with gilteritinib as monotherapy. Most patients had received front-line treatment with intensive chemotherapy, with approximately half receiving chemotherapy plus midostaurin (*n* = 67). Composite CR rates (25.4% and 27.5%) and median OS (6.4 and 7.8 months) were similar with prior midostaurin exposure or not and comparable to those observed in the ADMIRAL trial [8]. However, when compared with the results of the ADMIRAL trial, higher rates of grade ≥ 3 thrombocytopenia but equal rates of anemia were observed [21]. These findings support the use of gilteritinib, even in intensively treated patients who have received midostaurin as front-line therapy.

In order to reduce the rate of mortality and the utilization of healthcare resources, the United Kingdom National Health Service (NHS) made gilteritinib available as an emergency measure to patients aged > 16 y with R/R *FLT3* mutant AML starting from April 2020. A multicentric analysis in UK evaluated 50 R/R AML patients treated with gilteritinib; among them, most patients had previously received 1 (65%) or 2 (33%) lines of therapy, including intensive chemotherapy in a majority (86%). In total, 45% of patients had received a previous TKI inhibitor and 35% had relapsed after HSCT. A previous exposure to *FLT3* inhibitor (*p* > 0.9) and HSCT (*p* = 0.3) did not influence the median OS, which was 6.7 months. The composite CR/CR with incomplete hematological recovery (CRi) rate was 27%, and the mortality rate at day 30 and day 60 was 0% and 14%, respectively. Median time of hospitalization was 3.5 days in cycle 1, 0 days in cycles 2 and 3, and 1 day in cycle 4 [22].

The largest US multi-institutional retrospective analysis was recently reported. A total of 113 R/R AML patients were analyzed; most of them received gilteritinib as a single-agent therapy (62.8%), while the rest of the patients were treated with gilteritinib-based combinations (intensive chemotherapy (31%), hypomethylating agents 33%, venetoclax or hypomethylating and venetoclax 31%, and IDH inhibitors 5%). In total, 55 (48.7%) patients achieved a CRc, with CR in 25 patients (22.1%); the median OS was 7.0 months. A trend toward a higher CRc rate was observed in patients who received gilteritinib with combination treatments rather than as a single agent (64% vs. 43%, respectively, *p* = 0.09); however, no survival benefit was reported for combination therapy compared with the single-agent approach. The presence at baseline of *NRAS*, *KRAS*, and *PTPN11* mutations, which are known to confer gilteritinib resistance, was correlated with lower CRc (35% vs. 60.5%) and lower median OS than patients who did not express these mutations (4.9 months vs. 7.8 months; *p* < 0.01) [23].

Moreover, an Israeli group retrospectively analyzed 25 patients from six academic centers who received gilteritinib for *FLT3*-mutated R/R AML; most of them (80%) were treated with prior intensive chemotherapy and almost half (40%) with TKI therapy. The rate of CR was 48%, with an estimated OS of 8 months. Prior TKI exposure did not negatively impact OS and was associated with superior EFS (*p* = 0.016). The authors performed an age- and ELN-risk-matched comparison between patients who received gilteritinib and intensive salvage treatments. This analysis showed similar response rates (50% in both groups) and median OS (9.6 months vs. 7 months; *p* = 0.869) in the two groups, respectively [24]. Altogether, these studies showed comparable efficacy of gilteritinib in a real-life setting to the pivotal ADMIRAL trial (Table 2).

## 5. Safety Profile of Gilteritinib

In patients with R/R *FLT3*-mutated AML, gilteritinib exhibited an overall good safety profile. The integrated safety population (results from a phase I trial in Japanese patients [15], the phase I/II Chrysalis [17], and phase III ADMIRAL [8] studies) who received ≥1 dose of gilteritinib 120 mg (*n* = 319) is the main issue of this section. These patients were exposed to gilteritinib for an average time of 3.6 months. In total, 83.1% of patients experienced a treatment-related AE (TRAE) [9]. Anemia, febrile neutropenia, and thrombocytopenia were the most frequent grade 3 TRAEs observed in 60.2% of patients. In all, 33.9% of patients had serious TRAEs; the most common were febrile neutropenia, elevated alanine aminotransferase (ALT), and elevated aspartate aminotransferase (AST) levels [9]. A total of 6% and 29% of patients who received gilteritinib experienced dose reduction or stoppage due to an AE, respectively, while 7% of patients discontinued treatment due to an AE. Increased transaminases (51% of patients), myalgia or arthralgia (50%), fatigue or malaise (44%), fever (41%), mucositis (41%), oedema (40%), rash (36%), noninfectious diarrhea (35%), dyspnea (35%), and nausea (30%) were the most common nonhematological AEs of any grade (incidence 30%) [9]. The most frequent serious nonhematological AEs (incidence 5%) were fever (13%), dyspnea (9%), renal impairment (8%), elevated transaminases (6%), and noninfectious diarrhea (5%). Among 2% of those who received gilteritinib, there were fatal AEs: cardiac arrest (1%), differentiation syndrome (DS) (1%), and pancreatitis (1% each) [9]. Although rarely occurring, gilteritinib treatment resulted in a number of clinically severe AEs of particular interest (AESIs) [9]. In particular, DS is characterized by the release of the differentiation block, typical of AML blasts, which is the promotion of cell maturation that signals leukemic cells to extravasate from circulation into tissues in large numbers, causing tissue damage [25]. In the integrated safety population, DS appeared in 11 patients (3%) between days 2 and 75 after the start of treatment, regardless of the presence of leukocytosis. Most patients recovered after drug interruption. Grade 3 treatment-emergent posterior reversible encephalopathy condition (PRES) was observed in two patients (0.6%). Treatment-related QT prolongation was noted in 7.2% of patients, with 1.9% of those patients having significant QT prolongation. In 1.3% of patients, cardiac failure was deemed grade 3 and treatment-related (it was severe in 0.9% of patients). In total, 2.5% of patients developed a grade 3 treatment-related hypersensitivity responses, and 1.6% of those events were severe (inclusive of one patient who experienced anaphylaxis) (EMA). With the exception of elevated liver transaminases levels, which occurred more frequently in gilteritinib recipients, gilteritinib therapy and salvage chemotherapy in the ADMIRAL trial caused similar TEAEs in the first 30 days of treatment [18]. Except for cough (0.09 vs. 0.05 events per patient-year), increased AST level (1.26 vs. 0.76 events per patient-year), and increased ALT level (1.22 vs. 0.84 events per patient-year), the incidence of all exposure-adjusted TRAEs was lower in gilteritinib receivers than in salvage chemotherapy recipients. Receivers of gilteritinib experienced a frequency of 19.34 events per patient-year, and those receiving salvage chemotherapy experienced 42.44 events per patient-year of exposure-adjusted grade 3 TRAEs [17]. Gilteritinib had a stable safety profile beyond 2 years [18]. Real-life studies [21,22,23,24] have shown toxicities similar to those of clinical trials, further demonstrating the manageable toxicity profile of gilteritinib.

## 6. Combination Regimens Including Gilteritinib in R/R and De Novo AML

### 6.1. Gilteritinib Plus Azacitidine in FLT3-Mutated AML

Wang et al. [25] proposed a randomized phase 3 trial aimed to assess the efficacy and safety of gilteritinib plus azacitidine vs. azacitidine in newly diagnosed *FLT3*-mutated AML considered not eligible for intensive chemotherapy. Patients were randomized (2:1) to be treated with gilteritinib (120 mg/day orally) and azacitidine at standard dosage or azacitidine alone on a 28-day cycle. In all, 123 patients were enrolled, 74 included in the gilteritinib–azacitidine arm (median age, 78 years) and 49 in the azacitidine arm (median age 76 years); among them, 47.3% and 32.7% had an ECOG performance status (PS) of 2 in the two arms, respectively.

Authors found no significant difference in OS between the two arms; the median OS was 9.82 months and 8.87 months, respectively (HR 0.916; 95% CI, 0.529–1.585; *p* = 0.753). The median EFS was 0.03 months in both treatment arms; the CRc rate was significantly higher in the gilteritinib–azacitidine arm than in the azacitidine arm (58.1% and 26.5%, respectively; *p* < 0.001). Furthermore, authors observed a numeric improvement in OS with gilteritinib–azacitidine in some patient subgroups, but statistical significance was not reached. In the subgroup of patients stratified as having an ECOG PS of 0 to 1, the median OS was 13.17 months and 11.89 months, respectively (HR, 0.811; 95% CI, 0.409–1.608; *p* = 0.549); among patients with an *FLT3–ITD* allelic ratio of 0.5 or higher, the median OS was 10.68 months and 4.34 months, respectively (HR, 0.580; 95% CI, 0.285–1.182; *p* = 0.134). AE rates were similar between the arms. AEs of any grade occurred in 100% of patients in the gilteritinib–azacitidine arm and 95.7% of those in the azacitidine arm. The rate of grade 3 or higher AEs was 95.9% and 89.4%, respectively [26]. According to these data, this combination approach did not improve survival outcomes in patients, with newly diagnosed *FLT3*-mutated AML unfit for intensive treatment. Therefore, the trial was closed based on the protocol-specified boundary for futility and recommendations from the independent data monitoring committee.

### 6.2. Gilteritinib Plus Venetoclax in R/R AML

Venetoclax has been approved as a standard treatment in combination with low-dose cytarabine or hypomethylating agents for newly diagnosed AML ineligible for intensive chemotherapy [27,28]. Single-agent venetoclax showed limited activity in R/R AML [29]; however, in vitro reports demonstrated synergistic activity between venetoclax and *FLT3* inhibitors in preclinical models [30,31].

In an American, multicenter study, 61 patients with R/R AML, including 56 with *FLT3*-mutated disease, were enrolled to receive a combination regimen based on venetoclax and gilteritinib; 15 patients were enrolled in the dose-escalation phase and 46 were enrolled in the dose-expansion phase. The trial provided 400 mg of venetoclax once daily and gilteritinib at 80 mg or 120 mg once daily during dose escalation, with the recommended phase II dose being venetoclax at 400 mg and gilteritinib at 120 mg. Among the 56 patients with *FLT3*-mutated disease treated at any dose, after a median follow-up of 17.5 months, the modified composite CR (consisting of complete response, complete response with incomplete blood count recovery, complete response with incomplete platelet recovery, and morphologic leukemia-free state) rate was 75% (the CR rate was 18%). The median time to response and median remission duration was 0.9 months and 4.9 months, respectively, with a median OS of 10.0 months. Modified composite CR was observed in 14 (67%, CR in 29%) of 21 patients with no prior *FLT3* TKI exposure and in 28 (80%, CR in 11%) of 35 patients with prior TKI exposure. The median OS was 10.6 months and 9.6 months, respectively. Grade 3 or 4 AEs occurred in 97% of patients, mostly characterized by cytopenias (80%). AEs led to venetoclax and gilteritinib interruptions in 51% and 48% of patients and to discontinuation of treatment in 15% and 13%, respectively. Serious AEs occurred in 75% of patients, most commonly febrile neutropenia (44%) and pneumonia (13%) [32]. This combination approach produced a highly modified composite CR rate in patients with *FLT3*-mutated R/R AML; however, dose interruptions for cytopenias were very common, and this regimen showed a high toxicity profile.

The addiction of gilteritinib to azacitidine and venetoclax in *FLT3*-mutated AML was another fascinating triplet combination. In the phase I/II trial recently reported by Short et al., the ORR was 100% (27/27), with a 92% CR in newly diagnosed patients, a median OS that had not yet been attained, and an OS of 85% at 1 year. In R/R patients, the ORR was 70% (14/20), with a CR rate of 20% (4/20) and a median OS of 5.8 months. With a median OS of 10.5 months, outcomes were better in patients who had not previously received gilteritinib or venetoclax [33].

### 6.3. Gilteritinib Plus Chemotherapy in Patients with Newly Diagnosed AML

Recently, encouraging data on the association between gilteritinib and induction and consolidation chemotherapy were presented at the 10th Annual Meeting of the Society of Hematologic Oncology. Patients enrolled in this phase 1 trial (NCT02236013) were required to be at least 18 years of age with newly diagnosed AML and have an ECOG performance status of 2 or less; the presence of an *FLT3* mutation at baseline was not required. Dose escalation of gilteritinib was assessed in part 1 of the study to identify the MTD. Induction regimen provided 3 days of idarubicin with 7 days of cytarabine and 14 days of gilteritinib at doses of 20 mg, 40 mg, 80 mg, 120 mg, or 200 mg, given on days 4 through 17 for up to 2 cycles. The consolidation approach included high-dose cytarabine plus the same dose of gilteritinib given daily for the first 14 days of each cycle for up to 3 cycles. Finally, patients received maintenance treatment based on gilteritinib daily for 28 days for up to 26 cycles. The dose expansion study (part 2) provided gilteritinib at 120 mg a day, with induction, consolidation, and maintenance following the same treatment pattern as dose expansion trial. In part 3 of the study, the gilteritinib dosing schedule during induction was modified to begin with the completion of chemotherapy, running from days 8 through 21, and the other receiving 3 days of daunorubicin and 7 days of cytarabine. Consolidation and maintenance followed the same treatment pattern as parts 1 and 2. In part 4 of the study, gilteritinib was given up to 56 consecutive days during consolidation. A total of 79 patients were enrolled; among them, 56.4% of patients harbored *FLT3* mutations, 42.3% had *FLT3–ITD* mutations, and 41% had *FLT3*wt disease. At the end of treatment, the composite CR in patients with *FLT3* mutation was 90.9%, with 70.6% of patients achieving a CR. The 26-week, 1-year, and 2-year OS rates were 92.4%, 82.1%, and 69.2%, respectively, in this subgroup. Additional data showed that while censoring for HSCT, the median disease-free survival (DFS) for patients with *FLT3* mutations (*n* = 40) was 460 days (95% CI, 150–970), while the *FLT3*-negative population (*n* = 22) experienced a median DFS of 288 days (95% CI, 23–971). The MTD of gilteritinib was established to be 120 mg per day, and dose-limiting toxicities occurred in 15 of 78 (19.2%) patients given gilteritinib. AEs led to the discontinuation of gilteritinib in 24.4% of patients. Grade ≥ 3 treatment-emergent AEs were reported in 93.6% of patients [34]. According to these results, an effective antileukemic response was observed in terms of CR and OS, particularly in the *FLT3*-mutated subgroup in newly diagnosed AML who received gilteritinib in combination with intensive chemotherapy. These data support further trials to confirm the validity of this approach and to compare this regimen with the already approved treatment based on the combination of midostaurin with intensive chemotherapy in *FLT3*-mutated patients. Table 3 summarizes the trials including gilteritinib for the treatment of de novo AML. Table 4 summarizes the ongoing and recruiting studies including gilteritinib in combination with chemotherapy or other small molecules in R/R and de novo AML.

## 7. Maintenance Therapy with Gilteritinib after Allogenic Transplant

To date, there are currently no definitive results derived from randomized trials to validate the use of gilteritinib for post-HSCT maintenance therapy. The pivotal Astellas-sponsored MORPHO trial addressing the value of a gilteritinib maintenance therapy post-HSCT is currently ongoing, with results expected in 2025 (NCT02997202) [35]. However, recently, ASTELLAS announced that since relapse-free survival (RFS) was not statistically significant at the primary analysis, the study, including follow-up, will be stopped as per the study protocol (news provided by Astellas Pharma Inc., Tokyo, Japan; BMT CTN on March 2023). The BMT CTN 1506 is a randomized, phase III trial aimed to assess maintenance with gilteritinib vs. placebo after HSCT in patients with *FLT3–ITD*-mutated AML who achieved first CR (NCT02997202). Gilteritinib is given between days 30 to 90 after HSCT at 120 mg daily for 2 years. The study provides a deep-sequencing assay strongly sensitive to *FLT3–ITD* mutations for minimal residual disease testing, which will identify patients most likely to respond to the maintenance approach with gilteritinib [35,36].

In the ADMIRAL study, 20% (49 of 247) of patients enrolled in the gilteritinib arm and 10% (14 of 124) of patients treated with salvage chemotherapy were alive for ≥2 years. Among the patients still alive, 18 of 49 underwent HSCT and 16 continued gilteritinib as post-transplant maintenance treatment. Post-HSCT maintenance with gilteritinib resulted in improved OS and RFS, similar to the findings of prior studies with other *FLT3* inhibitors [18]. Among patients in the study receiving gilteritinib for maintenance, OS at 24 months was 96.2% compared with prior reports of 90.5% with sorafenib and 85% with midostaurin [37,38,39]. The RFS in the gilteritinib maintenance group was 89.7% at 24 months compared with prior reports of 85% with sorafenib and midostaurin [37,38,39]. Furthermore, several factors correlated with worsened graft-vs.-host DFS and RFS, including matched unrelated donor transplant, pretransplant antithymocyte globulin, and lack of maintenance *FLT3* inhibitors [18]. Recently, Perl et al. [39] reported data on patients included in the ADMIRAL study who underwent HSCT and received gilteritinib after transplantation as maintenance therapy. Patients in the gilteritinib arm proceeding to HSCT could receive post-transplantation maintenance with gilteritinib if they were within 30 to 90 days’ post-transplantation and had achieved CRc with effective engraftment and no post-transplantation complications. The OS rates at 12 and 24 months were 68% and 47%, respectively, for all transplant recipients. Even though there was a tendency for prolonged OS following pretransplant CRc, post-transplant survival was equivalent in the two arms. Following HSCT, patients who restarted gilteritinib showed low rates of pretransplantation CRc (20%) or CR (0%) recurrence. Increased ALT level (45%), pyrexia (43%), and diarrhea (40%), as well as grade 3 AEs, were the most frequently reported AEs with post-transplant gilteritinib. Grade 3 acute graft-vs.-host disease occurrences and associated mortality were infrequent. Overall, post-transplantation survival in the two study arms was comparable [40]. Recently the MD Anderson group reported the data of a retrospective analysis of adult patients with *FLT3–ITD* AML who underwent HSCT and thereafter received sorafenib or gilteritinib as post-transplant maintenance. A total of 55 patients were treated with either gilteritinib (*n* = 27) or sorafenib (*n* = 29); median time to initiation of gilteritinib was 60 days after transplant and median duration of time on gilteritinib was 385 days. The 1-year progression-free survival (PFS) (66% vs. 76%; *p* = 0.4) and relapse incidence (19% vs. 24%; *p* = 0.6) were similar between the two groups, respectively; the 1-year OS (78% vs. 83%; *p* = 0.4) was also comparable. However, nonrelapse mortality at 1 year was higher in the gilteritinib group (15% vs. 0%; *p* = 0.03) [41]. Moreover, the Japanese group retrospectively analyzed 25 *FLT3*-mutated R/R AML patients who received HSCT (14 patients received gilteritinib as maintenance therapy and 11 patients did not). The median time from transplant to the initiation of gilteritinib was 36 days, while the median starting dose was 40 mg (range 20–120 mg). Patients treated with gilteritinib showed significantly longer 1-year leukemia-free survival (100% vs. 36.4%; *p* = 0.0028) and 1-year OS (100% vs. 45.5%; *p* = 0.0075) than those without gilteritinib. Among patients showing positive minimal residual disease (MRD) or a noncomplete response before transplant (*n* = 19), those on gilteritinib maintenance showed a lower 1-year cumulative incidence of AML relapse (0% vs. 68.8%; *p* = 0.0028) [42]. These results support the hypothesis that gilteritinib maintenance therapy might prevent disease relapse after transplant, especially in patients with positive MRD at the time of HSCT.

## 8. Gilteritinib for Extramedullary AML Relapse

The *FLT3–ITD* gene mutation has been described to promote leukemic cell infiltration into visceral organs while inhibiting homing to the bone marrow by downregulation of *CXCR4* signaling [43]. Several studies have demonstrated that miRNAs may promote hematopoiesis and hematological diseases [42,43]. *FLT3*-mediated signaling controls the expression of several miRNAs, with both downregulation (miR-451 and miR-144) and upregulation (miR-155, miR-10a, and miR-10b) mechanisms. The expression of these small molecules seems to favor extramedullary blasts infiltration, although underlying mechanisms remain to be demonstrated [44]. Several case reports have described the efficacy of gilteritinib in patients with *FLT3* mutant extramedullary relapse before or after transplant. Perrone et al. first demonstrated the potential biological effect of gilteritinib within the central nervous system (CNS) [45]. Moreover, Vignal et al. reported the presence of gilteritinib in cerebrospinal fluid at therapeutic doses [46]. In another case [47], a patient experienced a right supraclavicular mass with simultaneously occurring AML blasts relapsed after ASCT in the bone marrow. Both extramedullary and medullary blasts presented *FLT3–ITD* mutation. Therapy with 120 mg/day of gilteritinib was started, determining a medullary and extramedullary CR and allowing the patient to proceed to a second HSCT [47]. Furthermore, gilteritinib seems to have efficacy also in infrequent localization of AML. Kim et al. [48] described a case of an *FLT3–ITD*-mutated patient who presented an AML relapse involving the temporal iris, ciliary body, and choroid by a leukemic infiltrative mass. The patient started treatment with oral gilteritinib, obtaining rapid regression of the tumor, with a significant improvement in visual acuity [48]. The mechanisms underlying the documented activity of gilteritinib in extramedullary AML are still unknown; however, this small molecule appears to hold efficacy in this setting and should be taken into consideration, especially in heavily pretreated patients.

## 9. Antifungal Prophylaxis in Patients Treated with Gilteritinib

Due to the fact that gilteritinib mainly undergoes CYP3A4-dependent metabolism, the manufacturer advises against using gilteritinib concurrently with drugs that strongly induce or inhibit CYP3A4 and instead suggests to select alternative treatments [10]. In the phase I/II CHRYSALIS trial, which examined possible drug–drug interactions between gilteritinib and moderate and strong CYP3A4 inhibitors (such as fluconazole, voriconazole, and posaconazole), gilteritinib exposure was found to be less than two times higher when an azole was also administered. The incidence of AEs did not vary between patients who received a moderate or strong CYP3A4 inhibitor and those who did not; hence, this increase was not deemed to be clinically relevant [17]. The effects of weak CYP3A4 inhibitors (such as itraconazole) and strong CYP3A4 inhibitors (such as fluconazole) on the pharmacokinetics of gilteritinib were assessed in an open-label drug–drug interaction research study. The findings showed that fluconazole was associated with a smaller increase in systemic exposure to gilteritinib (1.43-fold) compared with itraconazole (2.3-fold), which was linked with a significant increase in systemic exposure to gilteritinib [10]. The larger phase III ADMIRAL trial, however, forbade the use of posaconazole, itraconazole, and voriconazole, leaving unaddressed the issue on how to combine these drugs [8]. Aleissa et al. assessed the prevalence of AEs associated with gilteritinib in 47 patients who received gilteritinib either with or without antifungal triazoles. In the gilteritinib–triazole group, AEs related to gilteritinib were comparable to those in the gilteritinib group without triazole (75% vs. 55.5%, *p* = 0.23). The severity of AEs, dose reductions or discontinuations from gilteritinib (15% vs. 14.8%), and 90-day mortality (35% vs. 11.1%) were also comparable between the two groups [49]. However, how interactions between azoles and gilteritinib impact toxicities is not yet fully defined. Therefore, the European Hematology Association guideline on antifungal prophylaxis in patients with AML treated with novel-targeted therapies recommended triazole antifungal prophylaxis for patients who are heavily pretreated with gilteritinib [50].

## 10. Development of Resistances to Gilteritinib

Around 30% of patients who relapse after achieving a remission to type 1 *FLT3* inhibitors (midostaurin, gilteritinib, and crenolanib) carry mutations in the *RAS* pathway, making it the most prevalent mutation-derived mechanism of resistance to type 1 inhibitors. These mutations may appear as new mutations following therapy or as clonal proliferation with rising variant allele frequency (VAF) over the course of therapy [51]. Poorer outcomes in both primary and secondary relapse scenarios are linked to higher VAFs in *RAS/MAPK* mutations. *RAS* pathway mutations are less common with type 2 *FLT3* inhibitors (quizartinib) than with type 1 inhibition, occurring in just 6% of patients relapsing after type 2 inhibitors. *RAS*-mutated clones can spread in patients using quizartinib, even though *FLT3–TKD* mutations are the most common route of resistance to type 2 inhibitors [52]. It was hypothesized that the preservation of *FLT3* mutant clones can also depend on the bone marrow microenvironment (BMME). Indeed, soluble cytokines and growth factors together with cell–cell contact between leukemic cells and stromal cells within BMME can act as a mediator for the preservation of leukemic clones [53]. BMME adaptation and changes have been described alongside therapy. Patients relapsing after intensive chemotherapy courses were found to have considerably greater *FLT3* ligand levels, inducing AKT, ERK, and other proapoptotic proteins’ downregulation through *FLT3* ligand-*FLT3*wt binding. Despite *FLT3–ITD* inhibition, *FLT3*wt-mediated activation of these pathways promotes leukemic cell survival [54].

In the ADMIRAL study, 40 patients acquired new mutations during treatment. Among them, in 18 patients, the *RAS/MAPK* pathway was affected, while *FLT3* was involved in 6 cases (5 patients presented the F691L mutation); 3 had *WT1* (1 had the F691L mutation), 1 had *IDH1*, and 1 had *GATA2*. Thirteen patients (32.5%) had no new mutations. During relapse, *FLT3 F691L* gatekeeper mutations and mutations in the *RAS/MAPK* pathway genes were mutually exclusive [55]. *RAS/MAPK* and *FLT3 F691L* mutations were acquired by nontransplanted patients during relapse; however, the latter did not correlate with refractoriness. Uncertainty exists regarding the relationship between the dosage of gilteritinib and the prevalence of emergent *FLT3 F691L* gatekeeper mutations at relapse. In the ADMIRAL study, patients who received gilteritinib at 120 mg/day had a comparable incidence of *FLT3 F691L*, as seen in relapsed patients who received gilteritinib from 20 to 200 mg/day, but none of the patients receiving >200 mg/day acquired this kind of mutation at relapse. However, compared with other patients, those receiving 120 mg/day had improved OS [56]. Another study demonstrated a relationship between gilteritinib dose and occurrence of resistance in 22 *FLT3*-mutated patients analyzed at relapse by next-generation sequencing and single-cell analysis, reporting a more likely onset of *RAS* or *FLT3 F691L* mutations in those treated with doses below 200 mg [57].

Recently, it was reported that FF-10101, a selective and irreversible *FLT3* inhibitor, significantly inhibited *FLT3–ITD* and -TKD mutations, including *F691L* and *D835*, both in vitro and in vivo [58,59]. Fifty-two patients with R/R AML were enrolled in a phase I dose escalation study to test the inhibitor. In pretreated patients (median number of prior therapies, *n* = 3), continuous treatment with FF-10101 at a dose of 10–225 mg 4 times per day or 50–100 mg twice daily led to a composite CR rate of 13% and a partial response rate of 8%, including those with activating *FLT3–TKD* mutations resistant to gilteritinib and other *FLT3* TKIs. Well-tolerated doses of 50–75 mg twice daily resulted in long-lasting *FLT3* suppression. The trial is still ongoing but not recruiting patients [60].

Sitravatinib is a multikinase inhibitor under evaluation in ongoing clinical trials of several solid tumors. In a recent study, the antitumor activity of sitravatinib against *FLT3–ITD* and clinically relevant drug resistance in *FLT3* mutant AML were explored. The *FLT3–ITD*-*F691L* mutation caused resistance to gilteritinib and all other *FLT3* inhibitors, both in vitro and in vivo, whereas sitravatinib showed a potent inhibitory impact. With stronger and more consistent suppression of p-ERK and p-AKT than gilteritinib, sitravatinib maintained excellent efficacy against *FLT3* mutation in the presence of cytokines. Additionally, sitravatinib was more effective against patient blasts carrying *FLT3–ITD* in vitro and in the PDX model than gilteritinib [61].

## 11. Conclusions

Gilteritinib is an easy-to-use oral drug, with toxicities mainly represented by hematologic myelosuppression and high liver enzymes. Particular mention should be made of the promotion of differentiation of leukemic blasts in a sizeable subset of R/R *FLT3* patients [62], which has also been reported in patients treated with IDH-mutant AML treated with IDH inhibitors (enasidenib and ivosidenib), for the induction of QT prolongation, pancreatitis, embryo–fetal toxicity, and a rare neurologic complication: posterior reversible encephalopathy syndrome (PRES), which requires permanent discontinuation of the drug.

From a clinical point of view, gilteritinib has improved response and survival rates in comparison with different standard salvage chemotherapy regimens in the R/R AML setting. In the ADMIRAL trial, the median OS was almost double for gilteritinib (9 months vs. 5 months for standard chemotherapy). Indeed, gilteritinib represents a clinical upgrade, since patients with a primary refractory disease (i.e., refractory to standard induction and high-dose cytarabine) who carry an *FLT3* mutation are shifted to an oral drug that has fewer side effects and is more effective than conventional chemotherapy. This approach is so appealing that almost all patients with a relapsed AML are currently retested for *FLT3* mutations [63], even if patients acquiring an *FLT3* mutation at relapse represent a minority and its occurrence has been reported in less than 8% [64]. Moreover, the biology of *FLT3* mutation is complex: although always leading to an in-frame transcript, *FLT3–ITD* can vary in sequence and length (between 3 and >400 nucleotides), and despite the prognostic relevance of the allelic ratio, which corresponds to the size of the mutated clones carrying *FLT3–ITD* [65], there is no standardized cut-off value in the allelic ratio when prescribing (or not prescribing) gilteritinib to small clones. Indeed, patients with a high *FLT3–ITD* allelic ratio (≥0.77) showed a longer OS (7.1 vs. 4.3 months), and other comutations in *FLT3* molecular partner retain a prognostic impact [19]. However, searching for several comutations at relapse is currently unpractical, and methodological issues remain to be addressed regarding the standardization of the *FLT3–ITD* allelic ratio assay [63]. We summarized the results of different real-life studies of gilteritinib that confirm that patients treated in daily clinical practice attain results similar to patients randomized in the ADMIRAL trial [21,22,23,24].

At 2 years from the start of gilteritinib, only 26 (20%) patients survived in the ADMIRAL trial, and most of them (18) underwent HSCT as consolidation [18]. These data suggest that gilteritinib represents an excellent bridging therapy to allotransplant, and patients who continue gilteritinib often develop resistance by several mechanisms. As for the setting of maintenance after HSCT, the role of gilteritinib remains uncertain after the termination of the NCT02997202 trial [35]. More data are eagerly awaited to shed definitive light on this topic. Indeed, an intriguing question is raised by the clinical efficacy shown by sorafenib [66], which, albeit not active against *FLT3–TKD*, outperforms gilteritinib as maintenance in post-HSCT setting; however, a definitive answer could come only from an RCT comparing gilteritinib with sorafenib, which is improbable to be tested in the future. Unfortunately, the possibility of starting gilteritinib as a pre-emptive strategy only in patients who manifest a minimal residual disease positivity after HSCT (similarly to acute lymphoblastic leukemia Philadelphia-positive [67]) is hampered by technical difficulties to exactly quantify *FLT3* mutation [68]. Indeed, the consensus document from the European Leukemia Net minimal residual disease Working Party states that mutations in signaling pathway genes (*FLT3–ITD*, *FLT3–TKD*) most likely represent residual AML when detected but are often subclonal and have a low negative predictive value; these mutations are best used in combination with additional minimal residual disease markers [69].

The future developments of gilteritinib in the treatment of *FLT3*-mutated AML patients will also depend on the pending results of its association with other drugs. As reviewed, the combination with intensive chemotherapy for de novo AML is under study, but this field is already covered by midostaurin [37] and quizartinib [70], thus strongly limiting expectations for real innovation. The association of gilteritinib with hypomethylating agents (mainly with azacitidine) has been disappointing [26]. Finally, the combination with venetoclax produced modest improvement but at the cost of elevated hematological toxicity [32]. Conversely, the use of gilteritinib in maintenance after HSCT remains unproven given the early termination of the ongoing NCT02997202 trial [35], maybe halted by increased myelotoxicity after transplant or emergence of resistant clones in the setting of immunocompromised hosts.

In the meanwhile, we have gathered increased experience to deal with challenging presentations of AML, such as the extramedullary localization of myeloid sarcoma, where gilteritinib seems to have a role, as well as in patients presenting an invasive fungal infection. Nowadays, the most challenging issue in patients treated with gilteritinib remains how to overcome the occurrence of resistance. Resistance to TKI is common in several cancers and represents an evolutionary response to a selective pressure exerted at a subclonal disease level. In the gilteritinib arm, the median duration of CR was 23.0 months; the median durations of CRc and CR/CRh were 4.6 months and 10.0 months, respectively [18]; these data indicate that patients who achieve better hematological responses experience prolonged clinical benefit, while for resistant patients outcome is extremely poor.

We discussed the current state of the art of gilteritinib studies and evaluated the advantages and limitations of its use in R/R AML. Overall, these data are highly encouraging and open a new avenue to the further development of targeted therapy approaches in *FLT3*-mutated AML.

## Figures and Tables

**Figure 1 jcm-12-03647-f001:**
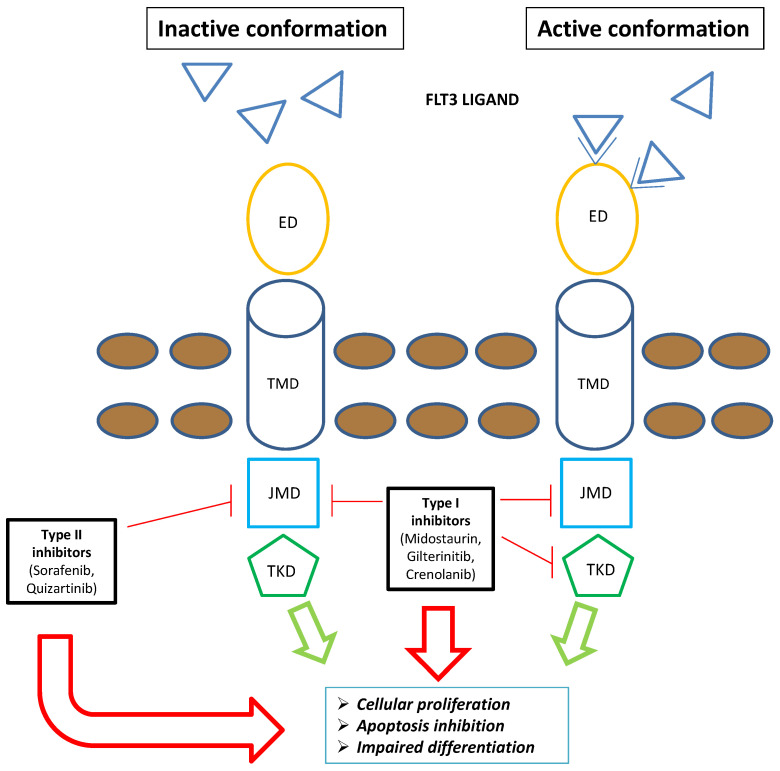
Schematic representation of *FLT3* inhibitors’ mechanism of action: The type I family of *FLT3* inhibitors (midostaurin, gilteritinib, and crenolanib) is able to bind the *FLT3* receptor both in the active and inactive conformation, inhibiting *FLT3*–ITD and TKD mutations. Contrarywise, the type II family of *FLT3* inhibitors (sorafenib and quizartinib) is able to bind the *FLT3* receptor in the inactive conformation, acting only on *FLT3–ITD*. Overall, *FLT3* inhibitors severely compromise leukemogenic activity of *FLT3* (i.e., cellular proliferation, apoptosis inhibition, and impaired differentiation). Blue triangle: FLT3 ligands. Brown circles: extracellular membrane. Green arrows: *FLT3*-mediated leukemogenic activity in the absence of *FLT3* inhibitors; red arrows: impairment of *FLT3*-mediated leukemogenic activity in the presence of *FLT3* inhibitors. Abbreviations: *FLT3*, FMS-like tyrosine kinase; TKD, tyrosine kinase domain; ED, extramembrane domain; TMD, transmembrane domain; JMD, juxtamembrane domain.

**Table 1 jcm-12-03647-t001:** Rate of responses and outcomes in ADMIRAL study.

Response Data	
Overall Response in *FLT3*-mutated AML ^1^ (*n* = 247)	
ORR ^2^ (%)	67.7
Complete remission or complete remission with partial hematologic recovery (%)	34
Complete remission (%)	21.1
No response (%)	26.7
Median duration of remission (months)	11
Rate of response in *FLT3*-mutated AML previously treated with TKIs ^7^ (*n* = 33)	
CRc ^3^ (%)	17
Median complete response duration (months)	8.9
CRc ^3^ (%) in AML who had previous midostaurin (*n* = 14)	57
CRc ^3^ (%) in AML who had previous sorafenib (*n* = 19)	47
Median complete response duration (months) in AML who had previous midostaurin	3.7
Median complete response duration (months) in AML who had previous sorafenib	12.9
Rate of response in *FLT3*-mutated AML ^1^ by baseline comutations (*n* = 239)	
CR ^4^/CRh ^5^ (%) in patients with DNA methylation/hydroxymethylation (*n* = 100)	29
CR ^4^/CRh ^5^ (%) in patients with transcription factors/regulators (*n* = 64)	17.2
CR ^4^/CRh ^5^ (%) in patients with chromatin–spliceosome–other (*n* = 47)	12.8
CR ^4^/CRh ^5^ (%) in patients with RTK–Ras signaling (*n* = 10)	20
CR ^4^/CRh ^5^ (%) in patients with *TP53*-aneuploidy (*n* = 15)	28.6
CR ^4^/CRh ^5^ (%) in patients with *NPM1* (*n* = 105)	27
CR ^4^/CRh ^5^ (%) in patients with *DNMT3A* (*n* = 75)	29.3
CR ^4^/CRh ^5^ (%) in patients with *DNMT3A/NPM1* (*n* = 55)	30.9
CR ^4^/CRh ^5^ (%) in patients with *WT1* (*n* = 45)	13.3
CR ^4^/CRh ^5^ (%) in patients with *IDH1/IDH2* (*n* = 38)	28.2
Outcomes data	
Outcomes of *FLT3*-mutated AML ^1^ (*n* = 247)	
Median OS ^6^ (months)	9.3
1-year OS ^6^ (%)	36.6
2-year OS ^6^ (%)	20.6
3-year OS ^6^ (%)	15.8
2-year cumulative relapse rate in patients who achieved a CR ^4^ (%)	52.6
2-year cumulative relapse rate in patients who achieved a CRc ^3^ (%)	75.7
Outcomes of *FLT3* mutated AML ^1^ according to previous TKIs ^7^ therapy	
OS ^6^ duration (months) in patients who did not receive prior TKIs ^7^ (*n* = 33)	9.5
OS ^6^ duration (months) in patients who receive prior TKIs ^7^ (*n* = 214)	8.7
Overall survival in *FLT3*-mutated AML by baseline co-mutations (*n* = 239)	
OS ^6^ (months) in patients with DNA methylation/hydroxymethylation (*n* = 100)	11.4
OS ^6^ (months) in patients with transcription factors/regulators (*n* = 64)	9.6
OS ^6^ (months) in patients with chromatin–spliceosome–other (*n* = 47)	7.1
OS ^6^ (months) in patients with RTK-Ras signaling (*n* = 10)	4.6
OS ^6^ (months) in patients with *TP53*-aneuploidy (*n* = 15)	10.6
OS ^6^ (months) in patients with *NPM1* (*n* = 105)	8.6
OS ^6^ (months) in patients with *DNMT3A* (*n* = 75)	11
OS ^6^ (months) in patients with *DNMT3A/NPM1* (*n* = 55)	15.1
OS ^6^ (months) in patients with *WT1* (*n* = 45)	8.3
OS ^6^ (months) in patients with *IDH1/IDH2* (*n* = 38)	15.4
Outcomes of *FLT3*-mutated AML ^1^ according *FLT3–ITD* length, multiple *FLT3–ITD* mutations, and *FLT3–ITD* allelic ratio	
OS ^6^ (months) in patients who had *FLT3–ITD* lengths > 51 bp (*n* = 90)	10.4
OS ^6^ (months) in patients who had *FLT3–ITD* lengths ≤ 51 bp (*n* = 99)	8.9
OS ^6^ (months) in patients with multiple *FLT3–ITD* mutations at baseline (*n* = 33)	9.3
OS ^6^ (months) in patients with high (≥0.77) *FLT3–ITD* allelic ratio (*n* = 109)	7.1
OS ^6^ (months) in patients with low (<0.77) *FLT3–ITD* allelic ratio (*n* = 113)	10.6

^1^ AML = acute myeloid leukemia. ^2^ ORR = overall response rate. ^3^ CRc = composite complete remission. ^4^ CR = complete remission. ^5^ CRh = complete remission with incomplete bone marrow recovery. ^6^ OS = overall survival. ^7^ TKIs = tyrosine kinase inhibitors.

**Table 2 jcm-12-03647-t002:** Real-life studies including gilteritinib as monotherapy in relapsed/refractory acute myeloid leukemia.

Reference	Number of Patients	Composite Complete Remission	Median Overall Survival	Comment
Dumas et al. [21]	140 (cohort B) 67 previously treated by intensive chemotherapy and midostaurin (cohort C)	25.4% (cohort B) 27.5% (cohort C)	6.4 months (cohort B) 7.8 months (cohort C)	Prognostic factors associated with OS identified female gender (HR 1.61), adverse cytogenetic risk (HR 2.52), and allogenic transplant after gilteritinib (HR 0.13)
Othman et al. [22]	50 (86% received previous intensive chemotherapy)	27%	6.7 months (95% CI 4.5—not reached)	The rate of composite complete response did not differ in those with previous exposure to *FLT3* inhibitors (23% vs. 32%, *p* = 0.6) or with past allogeneic transplant (29% vs. 27%, *p* = 0.3)
Numan et al. [23]	113 (62.8% received gilteritinib as monotherapy, while the remaining patients received gilteritinib in combination with other agents)	48.7%	7.4 months for transplant group 7.1 months for none-transplant 7.8 months in patients treated with prior midostaurin 5 months in patients treated with prior sorafenib	The presence of *PTPN11* and *NRAS* had a significant inferior impact on composite complete remission rate (59% vs. 37.5%) and median overall survival (4.9 months vs. 7.8 months; HR 2.4–95% CI 1.1–5.4 − *p* = 0.0057)
Shimony et al. [24]	25 (80% treated with prior intensive chemotherapy and 40% previously treated with tyrosine kinase inhibitor therapy)	48%	8 months	Prior tyrosine kinase inhibitor exposure did not negatively impact on overall survival and was associated with superior event-free survival (*p* = 0.016)

**Table 3 jcm-12-03647-t003:** Trials including gilteritinib for the treatment of de novo acute myeloid leukemia.

	Number of Patients	Median Age	Response	Mrdian Duration of Response	Median EFS ^1^/DFS ^2^	Survival	Number of Reference
5-Aazacitidine + Gilteritinib	74	78 years (range 59–90)	CRc ^3^ 58.1%	8.57 months	Median EFS ^1^ 4.53 months	Median OS ^5^ 9.82 months	[26]
5-Aazacitidine + Venetoclax + Gilteritinib	21	68 years (range, 18–82)	ORR ^4^ 100%	Not reported	Not reported	1-year OS ^5^ rate 85%	[33]
Standard chemotherapy (induction and consolidation) + Gilteritinb	44	50 years (range 23–77)	CRc ^3^ 90.9%	Not reported	Median DFS ^2^ 460 days	1-year OS ^5^ rate 82.1% 2-year OS ^5^ rate 69.2%	[34]

^1^ EFS = event-free survival. ^2^ DFS = disease-free survival. ^3^ CRc = composite complete remission. ^4^ ORR = overall response rate. ^5^ OS = overall survival.

**Table 4 jcm-12-03647-t004:** Ongoing and recruiting studies including gilteritinib.

Number of the Study	Protocol Regimen	Eligible Patients
NCT04027309	gilteritinib vs. midostaurin in combination with induction and consolidation therapy followed by one-year maintenance	newly diagnosed acute myeloid leukemia or myelodysplastic syndromes with excess blasts-2 with *FLT3* mutations
NCT04140487	azacitidine, venetoclax, and gilteritinib	relapsed/refractory *FLT3*-mutated acute myeloid leukemia, chronic myelomonocytic leukemia, or high-risk myelodysplastic syndrome/myeloproliferative neoplasm
NCT04240002	gilteritinib combined with chemotherapy	children, adolescents, and young adults with *FLT3–ITD*-positive relapsed/refractory acute myeloid leukemia
NCT05546580	iadademstat and gilteritinib	relapsed/refractory acute myeloid leukemia with *FLT3–ITD* mutation
NCT05520567	gilteritinib, venetoclax, and azacitidine	newly diagnosed with acute myeloid leukemia with *FLT3* mutations
NCT05028751	lanraplenib (lanra) in combination with gilteritinib	*FLT3*-mutated relapsed/refractory acute myeloid leukemia
NCT05010122	astx727, venetoclax, and gilteritinib	newly diagnosed, relapsed/refractory *FLT3*-mutated acute myeloid leukemia or high-risk myelodysplastic syndrome
NCT04293562	standard chemotherapy vs. therapy with cpx-351 and/or gilteritinib	newly diagnosed acute myeloid leukemia with or without *FLT3* mutations
NCT05010772	decitabine alone or in combination with venetoclax, gilteritinib, enasidenib, or ivosidenib as maintenance therapy	acute myeloid leukemia in remission

## Data Availability

The data presented in this study are openly available.

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
