# Peer review of "Gilteritinib: The Story of a Proceeding Success into Hard-to-Treat FLT3-Mutated AML Patients"

_jcm, 2023, doi:10.3390/jcm12113647_

Round 1

Reviewer 1 Report

The paper is well written and comprehensive.

Addition of more thorough review of pharmacokinetics and a comparison between Giltertinib and other TKIs approved for the same indication can be suggested.

The overall language can be more polished and simplified

Author Response

Reviewer 1

Addition of more thorough review of pharmacokinetics and a comparison between Giltertinib and other TKIs approved for the same indication can be suggested.

Reply: A section on pharmacokinetics of gilteritinib was added. To date, midostaurin is indicated in 1st line, while gilteritinib in ≥2 line. So we prefer not adding more.

Reviewer 2 Report

Well-written comprehensive review of the current status of Gilteritinib in AML. Important issues were adequately discussed about the interactions, side effects, extra medullary disease, resistance. New combinations were updated according to the recent publications, ongoing studies and official guidelines.  

No comments.

Author Response

we revised the English language

Reviewer 3 Report

The authors provide a comprehensive overview of the available data regarding gilteritinib in AML for relapsed/refractory patients, newly diagnosed patients, as monotherapy and in combination with chemotherapy in addition to other targeted therapies. 

Overall the manuscript would, in my opinion, benefit from the following:

1. Genes should be written italicized. For example FLT-3 ITD and FLT-3 TKD. 

2. Is gilteritinib equally effected in FLT-3 ITD and TKD mutations? This is not mentioned. It would also be useful to have a section on concurrent mutations with FLT-3 and efficacy of gilteritinib within these settings. This is touched on in Table 1.

3. The paragraphs in general are too long sometimes encompassing an entire page. This would be better if broken up. For example the section on the ADMIRAL trial is one paragraph while the section on side effects is also one paragraph. 

4. An unusual side effect of gilteritinib (unusual for most AML treatment but more commonplace with targeted therapy) is differentiation syndrome. It would be useful to include a brief explanation of differentiation syndrome in the manuscript.

5. Table 1 that focuses on the ADMIRAL trial has too much information like additional mutations. I think it would be more useful to have Table 1 be a summary of gilteritinib use in the relapsed/refractory setting and Table 2 to be a summary of frontline gilteritinib. 

6. The conclusion should be more definitive on the information determined from gilteritinib studies thus far and what the future holds. For example gilteritinib has clear frontline efficacy however the use of gilteritinib in maintenance is much more questionable. This is said throughout the manuscript in snippets and should be tied together in the conclusion including why this is so, ie resistance to the specific targeting by gilteritinib. 

7. The section on extramedullary disease is via case reports. Given the length of the review I would consider removal of this. 

8. The authors report that the efficacy of sorafenib is clearly better than gilterinib for maintenance in AML based on a systematic review. I would be careful with declarations like this- the review suggests that it is better. One would need a phase 3 randomized study to clearly show superiority. 

Author Response

Reviewer 3

  1. Genes should be written italicized. For example FLT-3 ITD and FLT-3
  2. Is gilteritinib equally effected in FLT-3 ITD and TKD mutations? This is not mentioned. It would also be useful to have a section on concurrent mutations with FLT-3 and efficacy of gilteritinib within these settings. This is touched on in Table 1.
  3. The paragraphs in general are too long sometimes encompassing an entire page. This would be better if broken up. For example the section on the ADMIRAL trial is one paragraph while the section on side effects is also one paragraph. 
  4. An unusual side effect of gilteritinib (unusual for most AML treatment but more commonplace with targeted therapy) is differentiation syndrome. It would be useful to include a brief explanation of differentiation syndrome in the manuscript.
  5. Table 1 that focuses on the ADMIRAL trial has too much information like additional mutations. I think it would be more useful to have Table 1 be a summary of gilteritinib use in the relapsed/refractory setting and Table 2 to be a summary of frontline gilteritinib. 
  6. The conclusion should be more definitive on the information determined from gilteritinib studies thus far and what the future holds. For example gilteritinib has clear frontline efficacy however the use of gilteritinib in maintenance is much more questionable. This is said throughout the manuscript in snippets and should be tied together in the conclusion including why this is so, ie resistance to the specific targeting by gilteritinib. 
  7. The section on extramedullary disease is via case reports. Given the length of the review I would consider removal of this. 
  8. The authors report that the efficacy of sorafenib is clearly better than gilterinib for maintenance in AML based on a systematic review. I would be careful with declarations like this- the review suggests that it is better. One would need a phase 3 randomized study to clearly show superiority. 

Reply: 

  1. It was done.
  2. It was already reported in lines 198-201, the better response in the ITD-high ratio sub-group.
  3. In the section of ADMIRAL trial, the aspect related to toxicity was redundant () and removed.
  4. Differentiation syndrome was explained briefly.
  5. We added a new table (table 3) which summarized trials including gilteritinib for the treatment of de novo AML
  6. 786-790 a sentence was added.
  7. This paragraph was considerably shortened.
  8. This phrase was mitigated and the need for a phase 3 study was reported
